# ASH1-catalyzed H3K36 methylation drives gene repression and marks H3K27me2/3-competent chromatin

Vincent T Bicocca[1]*, Tereza Ormsby[2], Keyur K Adhvaryu[3], Shinji Honda[4], Eric U Selker[1]*

[1]Institute of Molecular Biology, University of Oregon, Eugene, United States; [2]Department of Biochemistry Faculty of Science, Charles University, Prague, Czech Republic; [3]St. George's University School of Medicine, Grenada, Caribbean; [4]Faculty of Medical Sciences, University of Fukui, Fukui, Japan

**Abstract** Methylation of histone H3 at lysine 36 (H3K36me), a widely-distributed chromatin mark, largely results from association of the lysine methyltransferase (KMT) SET-2 with RNA polymerase II (RNAPII), but most eukaryotes also have additional H3K36me KMTs that act independently of RNAPII. These include the orthologs of ASH1, which are conserved in animals, plants, and fungi but whose function and control are poorly understood. We found that *Neurospora crassa* has just two H3K36 KMTs, ASH1 and SET-2, and were able to explore the function and distribution of each enzyme independently. While H3K36me deposited by SET-2 marks active genes, inactive genes are modified by ASH1 and its activity is critical for their repression. ASH1-marked chromatin can be further modified by methylation of H3K27, and ASH1 catalytic activity modulates the accumulation of H3K27me2/3 both positively and negatively. These findings provide new insight into ASH1 function, H3K27me2/3 establishment, and repression in facultative heterochromatin.

DOI: https://doi.org/10.7554/eLife.41497.001

*For correspondence:
bicocca@uoregon.edu (VTB);
selker@uoregon.edu (EUS)

**Competing interests:** The authors declare that no competing interests exist.

## Introduction

Methylation of histone H3 at lysine 36 (H3K36me) is largely associated with euchromatic regions of eukaryotic genomes (*Ho et al., 2014*). It serves as a link to transcription, as a H3K36 lysine methyltransferase (KMT; for example, yeast Set2) is directly associated with RNA polymerase II (RNAPII) elongation, and the mark is enriched along actively transcribed genes (*Kizer et al., 2005*; *Li et al., 2003*; *Morris et al., 2005*). As a result, H3K36me is commonly cited as an indicator of 'active' chromatin and is thought to exist in an antagonistic relationship with heterochromatin (*Gaydos et al., 2012*). Cohabitation of H3K36me3 with either H3K27me2/3 or H3K9me2/3 on the same histone tail is rare (*Jamieson et al., 2016*; *Voigt et al., 2012*; *Young et al., 2009*), and deposition of one mark can inhibit deposition of the second (*Schmitges et al., 2011*; *Voigt et al., 2012*; *Yuan et al., 2011*). Paradoxically, studies of H3K36me have shown that this modification can recruit chromatin remodelers and modifiers that organize and deacetylate nucleosomes, stabilize histones by inhibiting exchange, and restrict access to DNA – effectively conferring features of heterochromatin (*Carrozza et al., 2005*; *Fazzio et al., 2001*; *Lee et al., 2013*; *Li et al., 2007a*; *Smolle et al., 2012*). In metazoans, these seemingly dissonant functions are resolved by a division of labor within the H3K36me pathway that: 1) links conversion of H3K36me2 to –me3 with transcription elongation by physically tethering the Set2-ortholog to RNAPII, and 2) employs specialized RNAPII-independent KMTs to catalyze H3K36me1/2. The consequence is a complex and poorly understood regulatory network controlling access to and modification of the H3K36 substrate. Though much has been

**eLife digest** Not all genes in a cell's DNA are active all the time. There are several ways to control this activity. One is by altering how the DNA is packaged into cells. DNA strands are wrapped around proteins called histones to form nucleosomes. Nucleosomes can then be packed together tightly, to restrict access to the DNA at genes that are not active, or loosely to allow access to the DNA of active genes.

Chemical marks, such as methyl groups, can be attached to particular sites on histones to influence how they pack together. One important site for such marks is known as position 36 on histone H3, or H3K36 for short. Correctly adding methyl groups to this site is critical for normal development, and when this process goes wrong it can lead to diseases like cancer. An enzyme called SET-2 oversees the methylation of H3K36 in fungi, plants and animals. However, many species have several other enzymes that can also add methyl groups to H3K36, and their roles are less clear.

A type of fungus called *Neurospora crassa* contains just two enzymes that can add methyl groups to H3K36: SET-2, and another enzyme called ASH1. By performing experiments that inactivated SET-2 and ASH1 in this fungus, Bicocca et al. found that each enzyme works on a different set of genes. Genes in regions marked by SET-2 were accessible for the cell to use, while genes marked by ASH1 were inaccessible. ASH1 also affects whether a methyl group is added to another site on histone H3. This mark is important for controlling the activity of genes that are critical for development.

ASH1 is found in many other organisms, including humans. The results presented by Bicocca et al. could therefore be built upon to understand the more complicated systems for regulating H3K36 methylation in other species. From there, we can investigate how to intervene when things go wrong during developmental disorders and cancer.

DOI: https://doi.org/10.7554/eLife.41497.002

learned about H3K36me3 as a signal, there is little mechanistic understanding of how the RNAPII-independent KMTs are targeted and how their products function.

The complexity and significance of the H3K36me regulatory pathway is illustrated both in the range of fundamental genomic processes it underlies (e.g. transcription initiation and repression, alternative splicing, and DNA replication, recombination and repair) (*Wagner and Carpenter, 2012*), and the frequency with which it is disturbed during oncogenesis. The direct or indirect disruption of H3K36me by mutation of histone H3 genes defines distinct subtypes of pediatric chondroblastoma (H3.3K36M) and glioblastoma (H3.3G34R/V) (*Fang et al., 2016*; *Lu et al., 2016*; *Schwartzentruber et al., 2012*). In addition, recurrent mutation or overexpression of genes that methylate (Ash1L, Nsd1/2/3, and Setd2) or demethylate (Kdm2b and Kdm4a) H3K36 have been implicated as drivers of malignant transformation (*Black et al., 2013*; *He et al., 2011*; *Jaju et al., 2001*; *Kovac et al., 2015*; *Liu et al., 2012*; *Mar et al., 2014*; *Cancer Genome Atlas Network, 2015*). The prevalence of aberrant H3K36me regulation in cancer underscores the value of identifying therapeutic options for targeting this pathway. Unfortunately, the complexity and essential nature of the H3K36me pathway in higher organisms has restricted lines of inquiry and has left fundamental aspects of its function largely unexplored. Instead, much of the functional characterization of H3K36me has been performed in the yeasts *S. cerevisiae* and *S. pombe*, where H3K36me is non-essential and performed by a single RNAPII-associated KMT (*Strahl et al., 2002*). The simplicity of the H3K36me pathway in yeasts has proven valuable but has limited our understanding of the situation in eukaryotes that possess RNAPII-independent H3K36 KMTs, including filamentous fungi, plants, and animals (*Janevska et al., 2018*; *Schuettengruber et al., 2017*).

We present the filamentous fungus *Neurospora crassa* as an experimental bridge between yeasts and higher organisms, and use it to address unresolved questions concerning H3K36me. As in *S. cerevisiae* and *S. pombe*, H3K36me is not essential in *N. crassa* but unlike the case in the yeasts, we found that H3K36 methylation results from a division of labor between the RNAPII-associated SET-2 enzyme which can catalyze mono-, di-, and tri-methylation (*Adhvaryu et al., 2005*) and ASH1 (NCU01932). Notably, like higher organisms, Neurospora possesses both facultative heterochromatin – characterized by Polycomb Repressive Complex 2 (PRC2)-catalyzed H3K27me2/3

(*Jamieson et al., 2013*) – and constitutive heterochromatin – characterized by H3K9me3, HP1, DNA methylation and HDAC recruitment (*Freitag et al., 2004*; *Tamaru and Selker, 2001*; *Tamaru et al., 2003*). Both of these forms of heterochromatin are nonessential in Neurospora, facilitating studies of their interplay in vivo (*Jamieson et al., 2016*). The study presented here reveals a novel function for ASH1 and elucidates relationships between H3K36me, RNAPII, and facultative heterochromatin.

## Results

### ASH1 and SET-2 differentiate poorly- and robustly-transcribed genes

Our examination of the H3K36me pathway in Neurospora began with analyses of *set-2* and *ash1* mutant strains. In preliminary work, we found that, unlike *set-2*, which is dispensable for viability, *ash1* appears to be essential, as evidenced by the inability to generate a pure Δ*ash1* strain (*Colot et al., 2006*). Nevertheless, we found that we could build an *ash1* strain that should be catalytically inactive by mutation of Y888, which is required for coordinating the target lysine in SET protein superfamily members (*Figure 1A,B*) (*Dillon et al., 2005*). Strains harboring the *ash1*(Y888F) mutation displayed severely compromised growth but only minor reductions in global H3K36me2 and –me3 (*Figure 1C,D,E*). Δ*set-2* strains showed a dramatic loss of H3K36me2 but only minor impairment of growth (*Figure 1C,D,E*). We found that deletion of the 'Set2 Rpb1 Interacting' (SRI) domain of SET-2, which should decouple the enzyme from RNAPII (*Youdell et al., 2008*), resulted in a loss of H3K36me3 comparable to that seen with a *set-2* deletion, suggesting that RNAPII-associated SET-2 is responsible for nearly all H3K36me3 (*Figure 1G*). Weak H3K36me3 signals remain in each of these backgrounds, raising the possibility that ASH1 is responsible for some H3K36me3. Consistent with this possibility, *set-2; ash1*(Y888F) double mutants showed additive loss of the H3K36me2 observed in the single mutants and loss of the residual H3K36me3 signal (*Figure 1E,F, G*). This suggested ASH1 has weak H3K36me3 catalytic activity in vivo – a surprise given the in vitro activity of its orthologs (*An et al., 2011*; *Yuan et al., 2011*) and that the protein has a tyrosine at amino acid position 886 (*Figure 1B*), the predicted site of the 'Y/F-switch,' which is characteristic of SET domains in mono/di-KMTs (*Collins et al., 2005*).

As a step to identify the functions of *ash1* and *set-2*, we investigated the distribution of their activities across the genome. Our observation that the catalytic activity of ASH1 is not essential provided an opportunity to analyze separately the H3K36me2 and –me3 catalyzed by ASH1 and SET-2 by ChIP-seq in *set-2* knockout and *ash1*(Y888F) strains, respectively. Overall, we found that H3K36me2 and –me3 is associated with gene-rich DNA and excluded from constitutive heterochromatin, which is marked by DNA methylation and H3K9me3 (*Figure 2A*, *Figure 2—figure supplement 1A*). H3K36me2 catalyzed by ASH1 or SET-2 was found in distinct domains that apparently together produce the overall pattern of wildtype (WT) H3K36me2 (*Figure 2A,C*). We found ASH1-catalyzed H3K36me2 was prominent across the promoter and body of the genes that are silent or poorly transcribed in WT (*Figure 2B,C*, *Figure 2—figure supplement 1B*). Conversely, SET-2-catalyzed H3K36me2 was found associated predominantly with moderately- and highly-transcribed genes and was depleted from transcriptional start-sites (TSS) but enriched over gene bodies (*Figure 2B,C*). By this assay, SET-2 was found to mark most (>80%) genes, while ASH1 marked the minority (~20%) of genes that lacked SET-2-catalyzed H3K36me2 (p-value<$10^{-4}$) (*Figure 2B*, *Figure 2—figure supplement 1C*). In the *ash1* mutant, H3K36me3 was found restricted to sites of SET-2-catalyzed H3K36me2 (*Figure 2C*), and the intergenic H3K36me3 seen in WT was absent. Similarly, H3K36me3 at domains of ASH1-catalyzed H3K36me2 was lost in the *ash1* mutant. Consistent with the results of the western blot (*Figure 1E*), H3K36me3 was not entirely lost when *set-2* was deleted, and signal remained at regions with intense ASH1-catalyzed H3K36me2 (*Figure 2C*, *Figure 2—figure supplement 1D*).

### ASH1-catalyzed H3K36me maintains repression of poorly transcribed genes

We carried out RNAseq analyses to assess the effect of ASH1 and SET-2 activity on gene expression. We found that both *ash1* and *set-2* mutants have substantial, but distinct, changes in gene expression relative to WT. The *set-2* deletion showed a relatively symmetrical distribution of gene expression changes with 916 genes up-regulated and 1222 genes down-regulated (*Figure 3A*). In contrast,

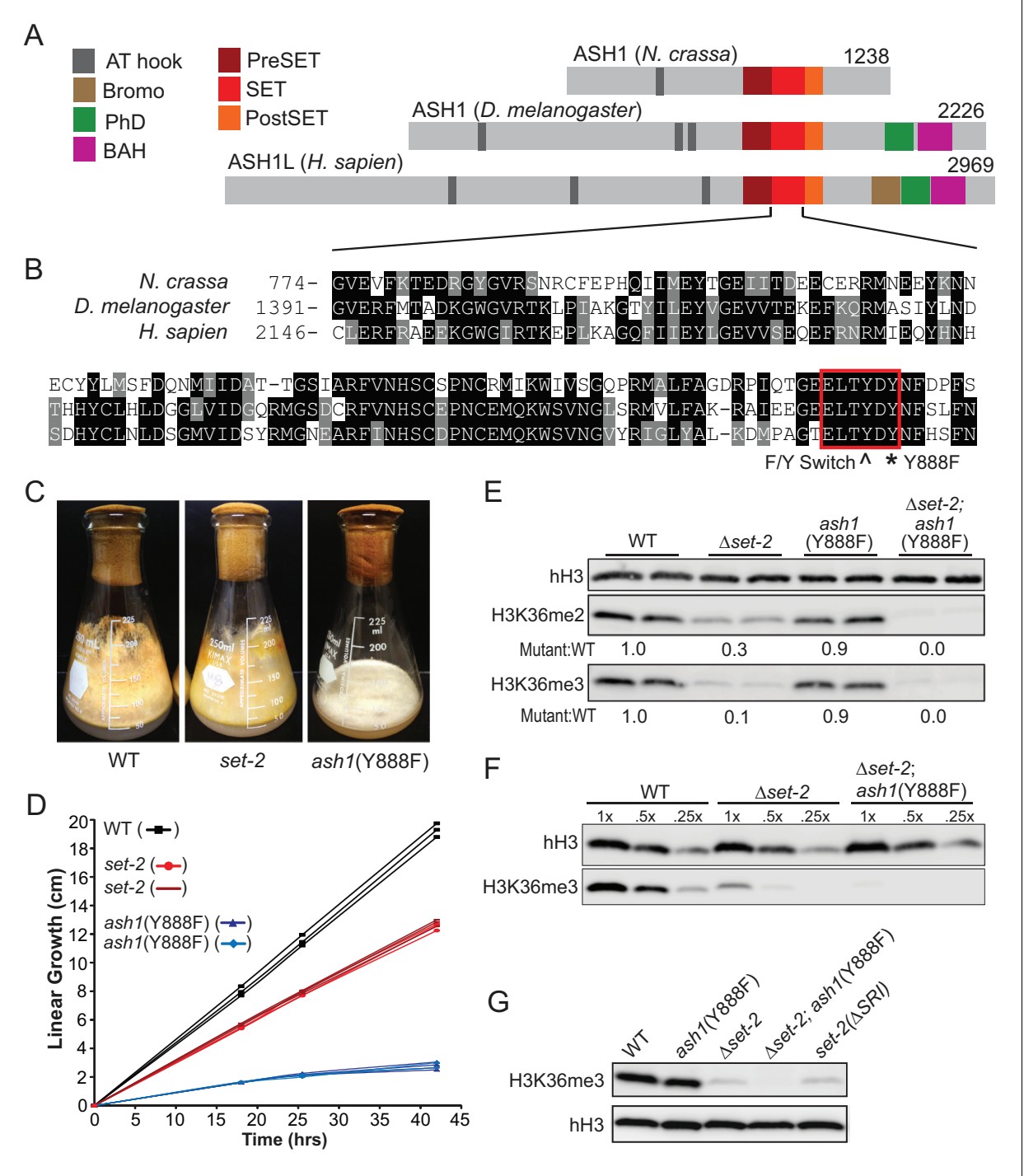

**Figure 1.** H3K36 KMT activity in *Neurospora crassa*. (**A**) Schematic of ASH1-orthologs in *Neurospora crassa*, *Drosophila melanogaster*, and *Homo sapiens*. (**B**) Multiple sequence alignment of the SET domain of ASH1 orthologs. Highlighted: F/Y Switch and Y888. (**C**) Culture flasks demonstrating growth phenotypes of Δ*set-2* and *ash1*(Y888F) strains compared to WT. (**D**) Linear growth rates of Δ*set-2*, *ash1*(Y888F), and WT. Biological replicates of the mutant strains were measured in triplicate. (**E**) Immunoblot analysis of H3K36me2 and H3K36me3 in WT, Δ*set-2*, *ash1*(Y888F), and Δ*set-2*; *ash1* (Y888F) backgrounds. Sibling replicates are included for each genotype. H3K36me signals were normalized to hH3 levels and compared to WT. (**F**) Immunoblot analysis of bulk H3K36me3 level in serial diluted extracts of WT, Δ*set-2*, and Δ*set-2*; *ash1*(Y888F) strains. (**G**) Immunoblot analysis of H3K36me3 levels in the *set-2* (ΔSRI) background.

DOI: https://doi.org/10.7554/eLife.41497.003

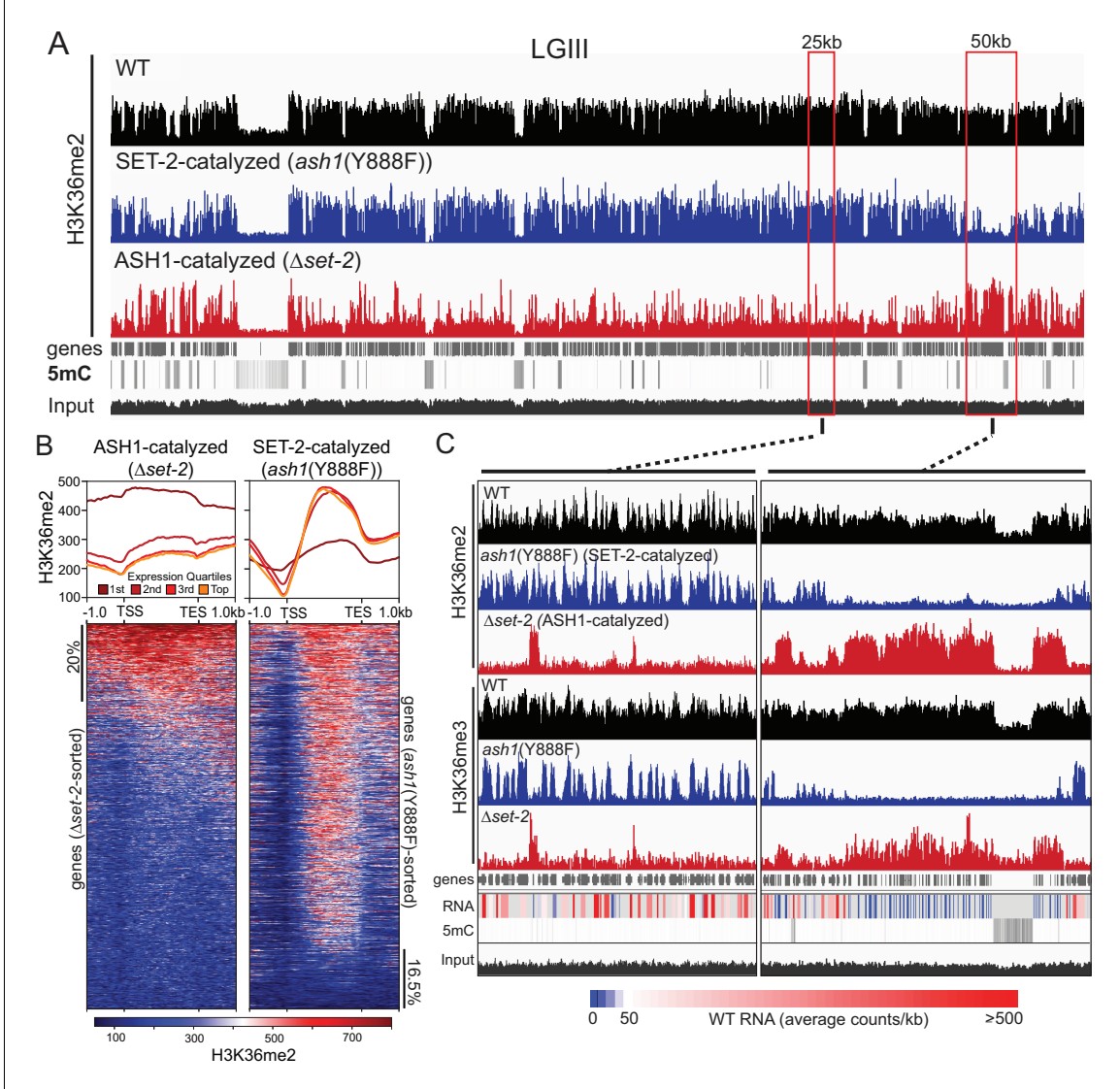

**Figure 2.** ASH1 and SET-2 specificities segregate the genome into compartments of poorly and robustly transcribed genes. (**A**) Representative IGV tracks of H3K36me2 ChIP-seq in WT, Δ*set-2*, and *ash1*(Y888F) backgrounds. Gene location, DNA methylation (to highlight constitutive heterochromatin), and 'input' tracks are included for reference. All of linkage group (LG) III is shown. (**B**) H3K36me2 profiles as determined by ChIP-seq in *set-2* and *ash1*(Y888F) backgrounds. Metaplots divide the H3K36me2 profile across gene quartiles determined by WT expression (i.e., '1 st'=genes in the lowest 25% of WT expression). Heatmaps were independently sorted by signal intensity in descending order. (**C**) IGV tracks of H3K36me2 and H3K36me3 ChIP-seq in WT, Δ*set-2*, and *ash1*(Y888F) backgrounds. Gene location, WT RNA abundance, DNA methylation, and input tracks are included for reference. Representative SET-2-rich and ASH1-rich regions are presented in the left and right panels, respectively.

DOI: https://doi.org/10.7554/eLife.41497.004

The following figure supplement is available for figure 2:

**Figure supplement 1.** (**A**) Representative IGV tracks of H3K36me3 ChIP-seq in WT, *ash1*(Y888F), and Δ*set-2* backgrounds are shown for LGIII.

DOI: https://doi.org/10.7554/eLife.41497.005

the *ash1* mutant predominantly resulted in up-regulation (1261 genes up-regulated; 228 genes down-regulated; *Figure 3A*). When we limited our analysis to ASH1-marked genes, they were almost exclusively up-regulated in the *ash1*(Y888F) background, while ASH1-unmarked genes showed no pattern of altered regulation (*Figure 3B*, *Figure 3—figure supplement 1*). When ASH1-marked genes were separated into 'SET-2-unmarked' and 'SET-2-comarked' categories, we found that co-marked genes were significantly up-regulated, while SET-2-unmarked genes showed little or no change in expression (*Figure 3C*). Collectively, these results imply that ASH1 and SET-2

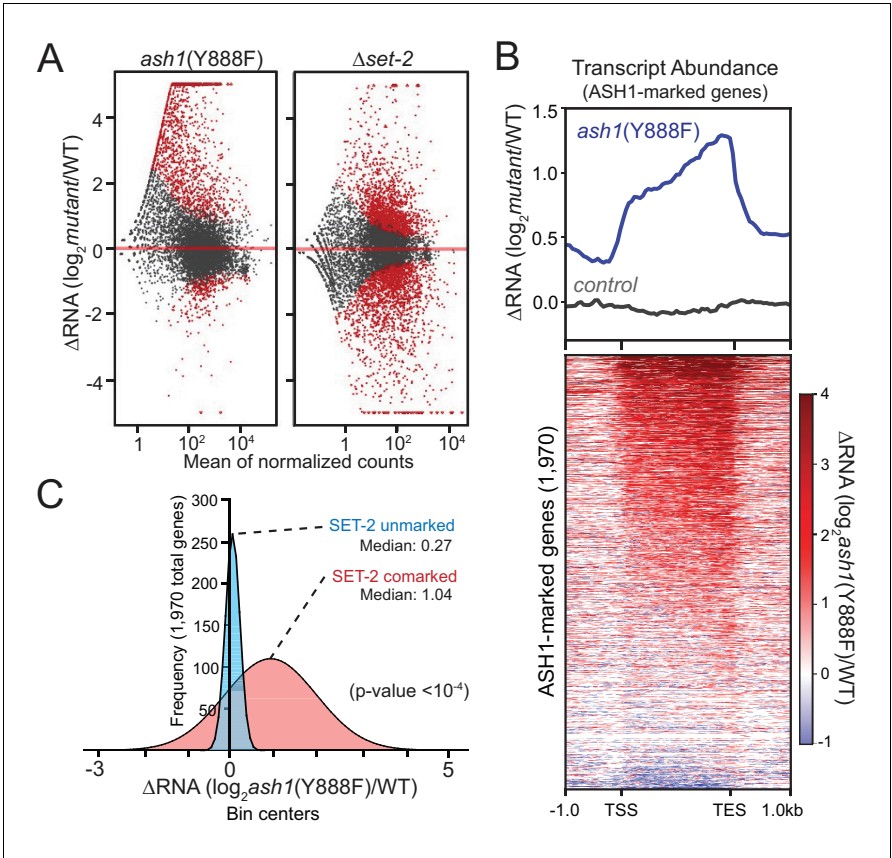

**Figure 3.** ASH1 catalytic activity maintains repression of poorly transcribed genes. (**A**) Gene expression changes are displayed as scatter plots of log2-fold changes *vs.* mean of normalized counts for *ash1*(Y888F) and *Δset-2* strains compared to WT controls. Duplicate biological replicates were analyzed, and points with p values < 0.1 are colored red. (**B**) Metaplot and heatmap illustrating change in RNA abundance as determined by RNAseq. *ash1* (Y888F) and WT replicates were normalized, averaged, and log2-ratios generated for ASH1-marked genes. The parent strain, N2930 (see *Materials and Methods*), is included as a control. (**C**) Frequency distribution of *ash1* (Y888F)/WT expression-change for genes marked by H3K36me2 in *Δset-2* strain ('ASH1-marked'; Guassian fit). SET-2-unmarked (blue) and SET-2-comarked (red) compartments are separated and median values highlighted. Statistical significance (two-tailed p-value<$10^{-4}$) was determined by a two sample Mann-Whitney test (Mann and Whitney, 1946).

DOI: https://doi.org/10.7554/eLife.41497.006

The following figure supplement is available for figure 3:

**Figure supplement 1.** (**A**) Average genic H3K36me2 levels (X-axis) catalyzed by ASH1 (defined in *Figure 2*) are plotted against change in gene expression (Y-axis) in the *ash1*(Y888F) background.

DOI: https://doi.org/10.7554/eLife.41497.007

independently catalyze H3K36me2 in a manner that differentiates the genome into regions of poorly- or robustly-transcribed genes. The repressed state of poorly transcribed genes is largely dependent upon ASH1 catalytic activity. Upon inactivation of ASH1, genes that are subject to derepression become co-marked by transcription-coupled SET-2.

## ASH1-catalyzed H3K36me delineates H3K27me2/3-competent chromatin

The presence of ASH1-catalyzed H3K36me at silent and poorly transcribed genes prompted us to investigate its relation to PRC2-catalyzed H3K27me2/3, which is also in domains of silent genes (*Jamieson et al., 2013*). Interestingly, we found nearly all (220/232) annotated domains of H3K27me2/3-marked chromatin (*Klocko et al., 2018*) are also marked with ASH1-catalyzed H3K36me2 (*Figure 4A*). When we looked at where the 12 absent domains were located, we saw

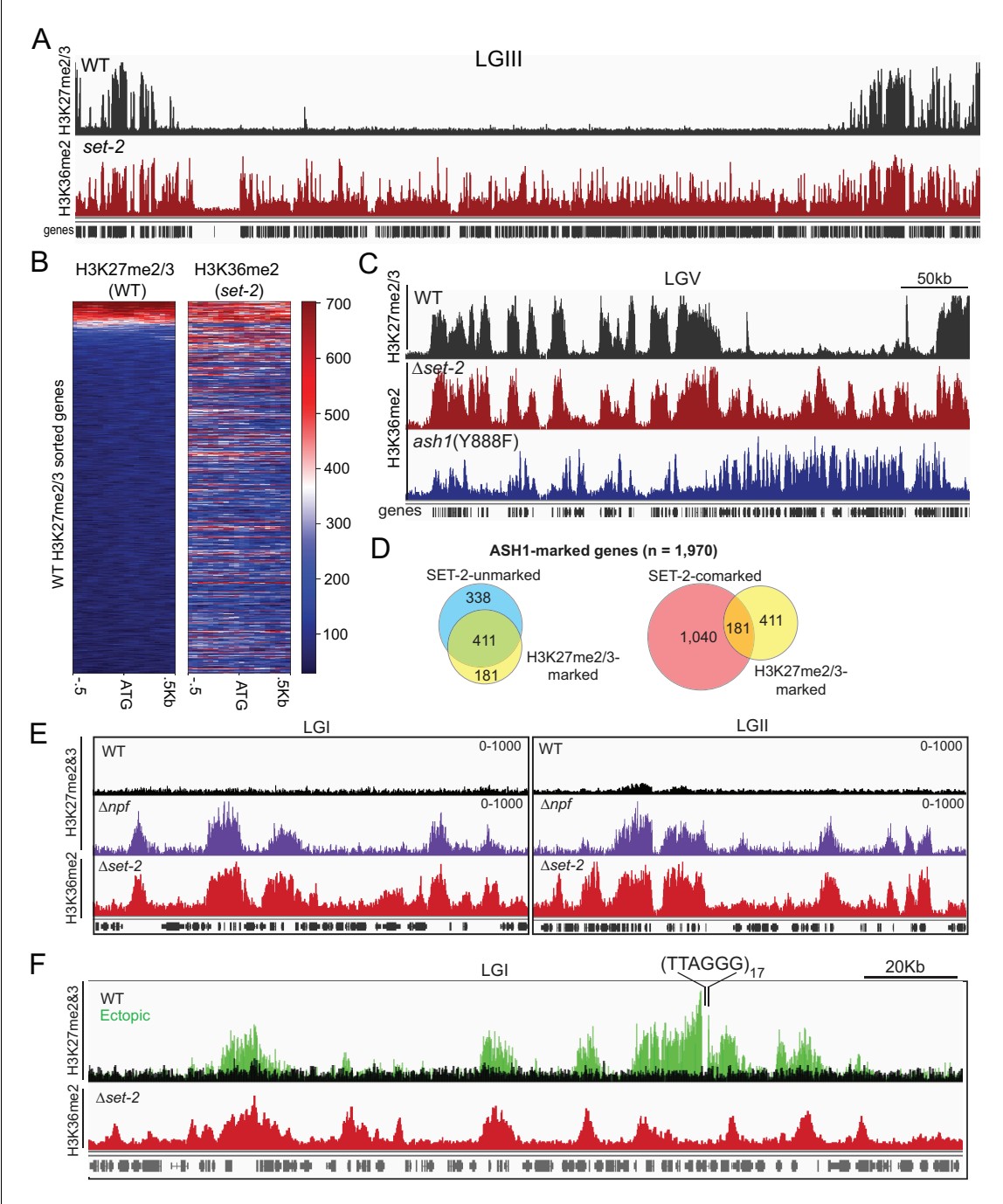

**Figure 4.** ASH1-catalyzed H3K36me2 delineates H3K27me2/3-competent chromatin. (**A**) Representative IGV tracks for H3K27me2/3 in WT and H3K36me2 as catalyzed by ASH1. All of LGIII is shown. (**B**) Heatmap showing the distribution of average H3K27me2/3 signal intensity in WT (left) and ASH1-catalyzed H3K36me2 (right) across the promoter region of all genes. Genes are sorted by WT H3K27me2/3 intensity. (**C**) Representative IGV tracks of H3K27me2/3 in WT and H3K36me2 as catalyzed by ASH1 or SET-2. (**D**) Fraction of ASH1-marked genes co-marked by SET-2, SET-7, or both SET-2 and SET-7. The distribution of SET-7/ASH1-comarked genes (yellow circle) in the SET-2-comarked (red) and SET-2-unmarked (blue) compartments shows that most (411/592) ASH1/SET-7 doubly marked genes are not marked by SET-2. Statistical significance (two-tailed p-value<10$^{-4}$) was determined by the Chi-square test. (**E**) H3K27me2/3 ChIPseq tracks from WT (black) and Δ*npf* (purple) strains are compared to H3K36me2 ChIPseq in Δ*set-2*. Depicted regions were selected for their multiple aberrant domains of H3K27me3. (**F**) H3K27me2/3 ChIPseq track from WT (black) and ectopic telomere-repeat (green) strains are superimposed and compared to H3K36me2 ChIPseq in a Δ*set-2* strain.

DOI: https://doi.org/10.7554/eLife.41497.008

they were all found in sub-telomere regions characterized by the presence of H3K27me2/3, H3K9me3, and DNA methylation (*Jamieson et al., 2018*), a finding consistent with ASH1-catalyzed H3K36me2 being excluded from constitutive heterochromatin (*Figure 2A*). When we examined the promoter region of individual genes, we again saw that the distribution of H3K27me2/3 overlapped with that of ASH1-catalyzed H3K36me2 (*Figure 4B*). Previous mass spectrometry analyses of *N. crassa* histone H3 suggested K27 and K36 methylation do not typically occur on the same molecule (*Jamieson et al., 2016*), implying that these marks exist as 'asymmetric' modifications on the same nucleosome and/or on adjacent nucleosomes (*Voigt et al., 2012*; *Yuan et al., 2011*). In all, we found 30% of ASH1-marked genes were co-marked by H3K27me2/3, and this co-marking was predominantly found at domains of ASH1-catalyzed H3K36me2 that lacked appreciable SET-2-catalyzed H3K36me2 (*Figure 4C,D*).

The consistent overlap of ASH1-catalyzed H3K36me with native H3K27me made us question whether the pattern would hold true in mutant backgrounds in which we had observed new domains of H3K27me. To test this, we first re-examined the H3K27me2/3-defects caused by deletion of the Drosophila Nurf55/Caf1 ortholog, Neurospora p55 (NPF) (*Jamieson et al., 2013*). Though the predominant effect of *npf* deletion is loss of sub-telomeric H3K27me2/3 (*Jamieson et al., 2013*), we also found new domains of H3K27me3 and, interestingly, these were limited to regions of ASH1-catalyzed H3K36me2 (*Figure 4E*). Next, we took advantage of a situation in which H3K27me2/3 was induced in a normally euchromatic region by insertion of telomere repeats in the vicinity (*Jamieson et al., 2018*). Using an insertion at the *csr-1* locus, we found that the discontinuous spread of H3K27me2/3 from the repeats correlated perfectly with the presence of ASH1-catalyzed H3K36me2 (*Figure 4F*). Altogether, these analyses show that a fraction of ASH1-marked chromatin is asymmetrically modified by PRC2 to generate overlapping profiles of H3K27me and H3K36me at genes that are most refractory to derepression, and that H3K27me-competency is a distinguishing characteristic of ASH1-marked chromatin.

## ASH1 activity influences H3K27me2/3 accumulation

Drosophila Ash1 has previously been reported to inhibit PRC2-mediated repression by preventing H3K27me2/3 accumulation (*Papp and Müller, 2006*). Similarly, H3K36me3 has been shown to inhibit catalysis of H3K27me2/3 in vitro (*Yuan et al., 2011*). We therefore asked if ASH1-catalyzed H3K36me is influenced by loss of H3K27me, and if H3K27me is influenced by loss of ASH1 activity. Immunoblotting and ChIPseq in a double-mutant strain lacking SET-2 and the H3K27 KMT, SET-7, revealed ASH1-catalyzed H3K36me2 was unchanged by loss of H3K27me2/3 across the genome (*Figure 5—figure supplement 1*), indicating that ASH1-catalyzed H3K36me2 is not dependent on PRC2-catalyzed H3K27me2/3.

We next asked whether some fraction of normal H3K27me regions depend on H3K36 methylation directed by ASH1. To test this possibility, we performed H3K27me2/3 ChIP in the *ash1*(Y888F) background. Inactivation of ASH1 showed a striking effect on H3K27me2/3, resulting in reduction or complete loss of the mark across roughly one-third of ASH1/PRC2-comarked genes (*Figure 5A,B*). The loss of ASH1-catalyzed H3K36me and resultant loss of H3K27me2/3 was accompanied by accumulation of H3K27 acetylation (ac) and derepression of affected genes (*Figure 5C,D*, *Figure 5—figure supplement 2*).

In addition to identifying ASH1-dependent H3K27me2/3, we also found domains of H3K27me2/3-competent chromatin where the ASH1 mark prevented H3K27me2/3. In the *ash1* mutant, 128 genes gain H3K27me2/3 (defined as >2 fold increase over background) (*Figure 5E*), whereas 180 genes lost the H3K27me2/3 mark (*Figure 5B*). Importantly, these new domains of PRC2-marked chromatin in the *ash1*(Y888F) strain are delineated by regions normally marked with ASH1-catalyzed H3K36me2 (*Figure 5F*). Thus, ASH1 catalyzed H3K36me can both positively and negatively influence H3K27me2/3 accumulation.

## Discussion

With the notable exception of yeasts, the H3K36me pathway of eukaryotes is divided between Set2 orthologs (SET2D in humans), which can catalyze mono-, di-, and tri-methylation, and a group of specialized KMTs that largely catalyze mono/di-methylation (*Wagner and Carpenter, 2012*). Study of the functional relationships between Set2-orthologs and the mono/di-KMTs has been limited, in part

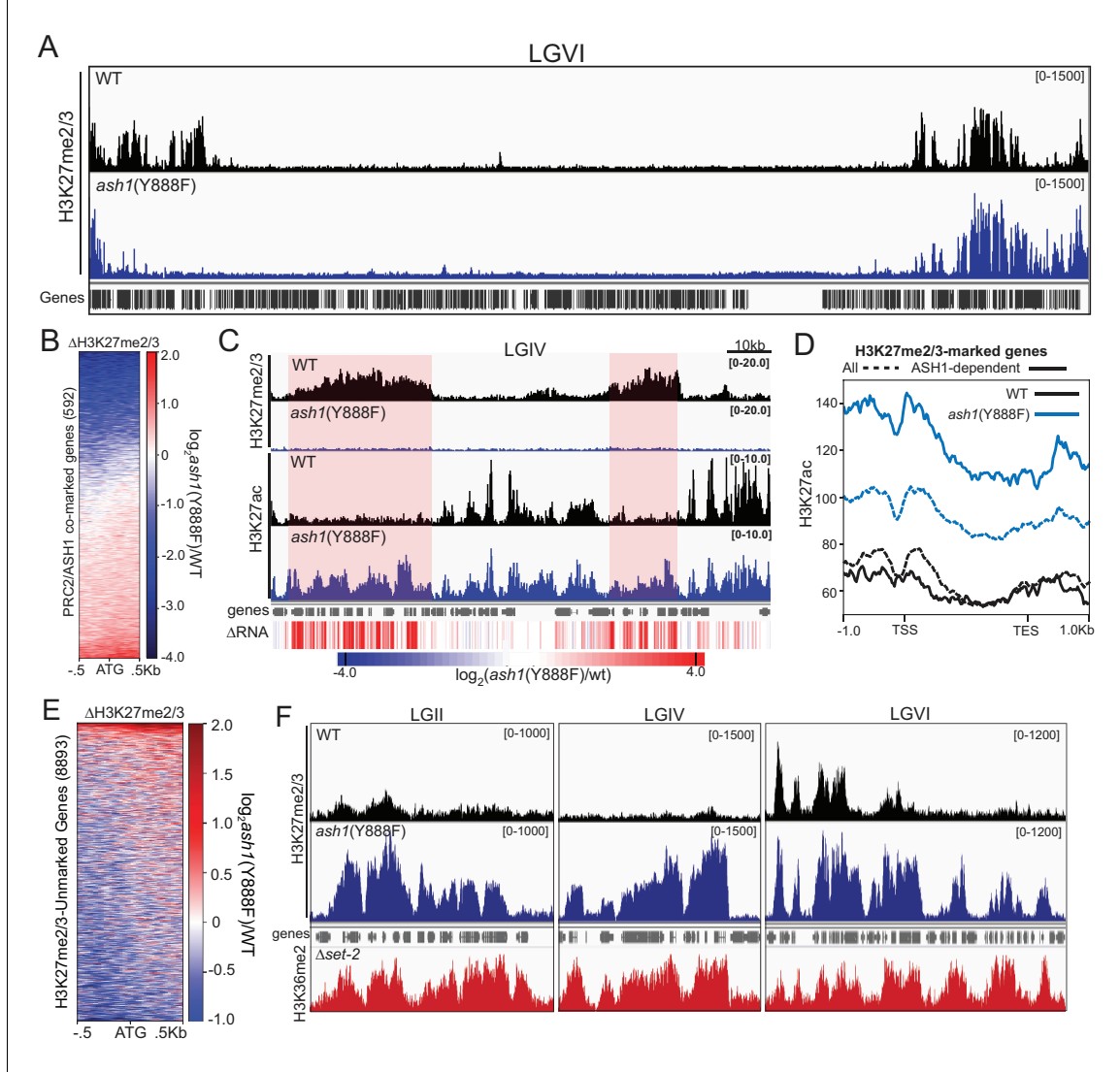

**Figure 5.** ASH1 activity differentially regulates H3K27me2/3 accumulation. (**A**) Representative IGV tracks of H3K27me2/3 in WT and *ash1*(Y888F) are shown for LGVI. Gene locations are included for reference. (**B**) Heatmap highlighting frequency and intensity of H3K27me2/3 loss over in the *ash1*(Y888F) background. (**C**) IGV tracks of H3K27me2/3 and H3K27ac in WT and *ash1*(Y888F) strains. Regions of H3K27ac accumulation correlating with gene upregulation (ΔRNA track) are highlighted. (**D**) Metaplot of H3K27ac accumulation in WT (black) and *ash1*(Y888F) (blue) strains at either all WT H3K27me2/3-marked genes (dashed line) or genes identified as 'ASH1-dependent' (solid line). (**E**) Change in H3K27me2/3 signal intensity by ChIPseq in the *ash1*(Y888F) background. Only genes established as 'unmarked' are included (*Figure 4B*). (**F**) ChIPseq tracks demonstrating H3K27me2/3 gains in *ash1*(Y888F) and comparison to H3K36me2 in the *set-2* background. Depicted regions were selected for their multiple aberrant domains of H3K27me3.

DOI: https://doi.org/10.7554/eLife.41497.009

The following source data and figure supplements are available for figure 5:

**Figure supplement 1.** (A, B) H3K36me2 ChIPseq in a *set-2; set-7* strain displayed on IGV with WT H3K27me2/3 and *set-2* H3K36me2.

DOI: https://doi.org/10.7554/eLife.41497.010

**Figure supplement 2.** Gene expression changes are summarized for genes marked by ASH1-catalyzed H3K36me2 (Top) and genes comarked by ASH1-catalyzed H3K36me2 and SET-7-catalyzed H3K27me2/3 (Bottom) in the *ash1*(Y888F) and Δ*set-7* strains.

DOI: https://doi.org/10.7554/eLife.41497.011

**Figure supplement 2—Source data 1 .** Gene expression changes (log2 *ash1*(Y888F)/wt).

DOI: https://doi.org/10.7554/eLife.41497.012

because numerous dedicated H3K36 mono/di-KMTs are found in higher organisms (e.g. seven in mammals) and these enzymes – as well as the Set2-ortholog – are typically essential, making determination of their independent actions difficult or impossible (*Wagner and Carpenter, 2012*). To address unresolved questions regarding the functional relationship between H3K36 KMTs, we took advantage of the simplified H3K36me pathway of *N. crassa*, which we showed consists of two H3K36 KMTs, SET-2 and ASH1. Unlike SET-2 (*Adhvaryu et al., 2005*), ASH1 appears to be essential for viability but we found that the organism tolerates a catalytic null mutation of *ash1*, allowing us to assess the relative contribution of the two KMTs. By dissecting the roles of these enzymes, we uncovered a previously undescribed pathway that connects ASH1-catalyzed H3K36me to repression of poorly transcribed genes. Curiously, we found that much of ASH1-marked chromatin is characterized by H3K27me2/3-competency. Not only did native domains of H3K27me overlap with domains of ASH1-catalyzed H3K36me, but experimentally-induced domains of H3K27me2/3 selectively spread to ASH1-marked chromatin.

RNAPII-associated SET-2 is generally considered to be the only KMT capable of catalyzing H3K36me3 (*Adhvaryu et al., 2005*), and based upon biochemical studies with Drosophila and human orthologs (*An et al., 2011*; *Yuan et al., 2011*) we expected ASH1 to act as a dedicated H3K36 mono/di-KMT. Conservation of a tyrosine residue at the 'F/Y-switch' of its SET domain (Figure S1A) supported this expectation (*Collins et al., 2005*). Results of western blotting suggested, however, that ASH1 is responsible for ~5% of global H3K36me3 (*Figure 1*) in the absence of SET-2, and the ASH1 homologs of *Fusarium fugikuroi* and *Plasmodium falciparum* have also been reported to have H3K36me3 activity (*Janevska et al., 2018*; *Jiang et al., 2013*). Although we detected ASH1-catalyzed H3K36me3 in vivo with different, independently validated, antibodies and techniques, it remains possible these antibodies recognized residual H3K36me2 or that ASH1 can convert H3K36me2 to –me3 but only in the absence of SET-2.

Neurospora SET-2 catalyzes H3K36me2/3 across the bodies of active genes, much as in yeast (*Krogan et al., 2003*; *Li et al., 2003*), whereas ASH1 is responsible H3K36me2/3 across large domains that encompass multiple genes and intergenic regions. Interestingly, genes marked by ASH1 are silent or poorly transcribed and are largely reliant upon the mark for their repressed state. Genes that were derepressed by inactivation of ASH1 were mostly 'SET-2-comarked,' that is, they only lost H3K36me in the absence of both KMTs. It will be interesting to learn how ASH1 is directed to where it acts, that is, in domains of lowly transcribed genes. Neurospora ASH1 does not display telling conserved protein domains, but does have an AT-hook that might interact with the minor-groove of A/T-rich DNA. Constitutive heterochromatin in Neurospora is characterized by A/T-rich DNA (*Cambareri et al., 1989*), but we found no indication that ASH1 functions at constitutive heterochromatin; in fact, H3K36me appears to be normally excluded from such regions.

Our finding that ASH1 has a function in repression is not entirely surprising given prior evidence of H3K36me3 in recruiting repressive chromatin machinery (*Fazzio et al., 2001*; *Keogh et al., 2005*; *Strahl et al., 2002*) but it was striking to see the extent of its repressive influence. When compared to PRC2-catalyzed H3K27me2/3 (*Jamieson et al., 2013*), ASH1-catalyzed H3K36me appears to be the predominant repressive modification of poorly transcribed genes (H3K27me covers only ~30% of silent, ASH1-marked, genes). Given the collaborative relationship between H3K36 KMTs, chromatin remodelers, and histone deacetylases (HDACs) described in other organisms (*Lee et al., 2013*), we predict a role for nucleosome positioning and histone deacetylation in ASH1-mediated repression. Here, we observed an accumulation of H3K27ac following loss of ASH1-dependent H3K27me2/3, but it is unclear if this is a passive product of H3K27me2/3 loss or if there is an active role for H3K27me2/3 in exclusion.

Future studies should examine the Neurospora counterpart of the yeast HDAC complex, RPD3 Small (RPD3S), which includes the H3K36me3 reader, EAF-3 (*Joshi and Struhl, 2005*; *Keogh et al., 2005*). RPD3S activity appears to be dependent upon proper nucleosome spacing established by Isw2 and Chd1, which together apparently organize and stabilize nucleosomes to restrict internal initiation by RNAPII in the wake of transcription (*Carrozza et al., 2005*; *Fazzio et al., 2001*; *Lee et al., 2013*; *Li et al., 2007a*; *Smolle et al., 2012*). Importantly, this mechanism is dependent upon transcription, as H3K36me3 deposition by SET-2 is strictly tied to elongating RNAPII (*Youdell et al., 2008*). Our results support a related transcription-independent mechanism that maintains gene repression at facultative heterochromatin. ASH1 would establish the H3K36me mark required to recruit RPD3S, while chromatin remodelers would establish proper nucleosome positions to facilitate

RPD3S deacetylase activity. To test this hypothesis, further study of the RPD3S HDAC will be required, but it will be challenging as the *N. crassa* ortholog of Rpd3 (HDA-3) is essential, and other units of the complex – EAF-3, SIN3, and NPF – are components of various other chromatin modifying complexes (*Jamieson et al., 2013*; *Sathianathan et al., 2016*). Notably, orthologs of EAF-3 and NPF (Mrg15 and Nurf55, respectively) have recently been identified as Ash-1 complex members in Drosophila (*Huang et al., 2017*; *Schmähling et al., 2018*), further supporting a connection to RPD3.

Perhaps the most surprising observation from our study was the substantial overlap of H3K27me2/3 at ASH1-marked chromatin. Neurospora H3K27me2/3 is catalyzed by a PRC2 complex that is highly similar to those found in metazoans (*Jamieson et al., 2013*) but *N. crassa* has no apparent PRC1 components. Even in higher organisms, the mechanism of repression mediated by PRC2 and H3K27me2/3 is far from clear, necessitating additional studies. Interestingly, we observed in Neurospora that loss of ASH1-dependent H3K36 methylation was associated with both losses and gains of H3K27me2/3. Early work with Drosophila gave evidence that ASH1 opposes the action of PRC2 function (*Klymenko and Müller, 2004*; *Shearn, 1989*), consistent with the observation that the presence of H3K36me on a histone tail can inhibit PRC2 activity in cis (*Yuan et al., 2011*), but our findings suggest the situation is more complicated. We found new domains of H3K27me2/3 at ASH1-regulated regions when ASH1 was inactivated, suggesting the presence of the ASH1 mark prevents H3K27me2/3. In addition, we found that ASH1 drives repression, and derepression associated with ASH1 inactivation is frequently accompanied by H3K27me2/3 loss. These seemingly opposing activities may reflect differential histone modifications and accompanying effector proteins found at those regions. Or perhaps, similar to plants, different forms of PRC2 may exist that respond differently to the presence of H3K36me (*Schmitges et al., 2011*). These possibilities will be interesting to investigate in the future. Finally, it is important to note that though it was initially surprising to find genome-wide colocalization of H3K27me2/3 with ASH1-catalyzed H3K36me2, this does not appear to be unique to Neurospora and other fungi, as recent work with embryonic stem cells revealed apparent cross-talk of these marks (*Streubel et al., 2018*).

Our work supports a model in which the genes of *N. crassa* are separated into two compartments depending upon their source of H3K36me (*Figure 6*). Actively transcribed genes possess SET-2-catalyzed H3K36me2/3 specific to the gene body, while silent and infrequently transcribed genes are covered in large domains of ASH1-catalyzed H3K36me2/3. In both cases, H3K36me appears to act as a repressive mark, protecting active genes against internal cryptic-transcription (*Li et al., 2007b*)

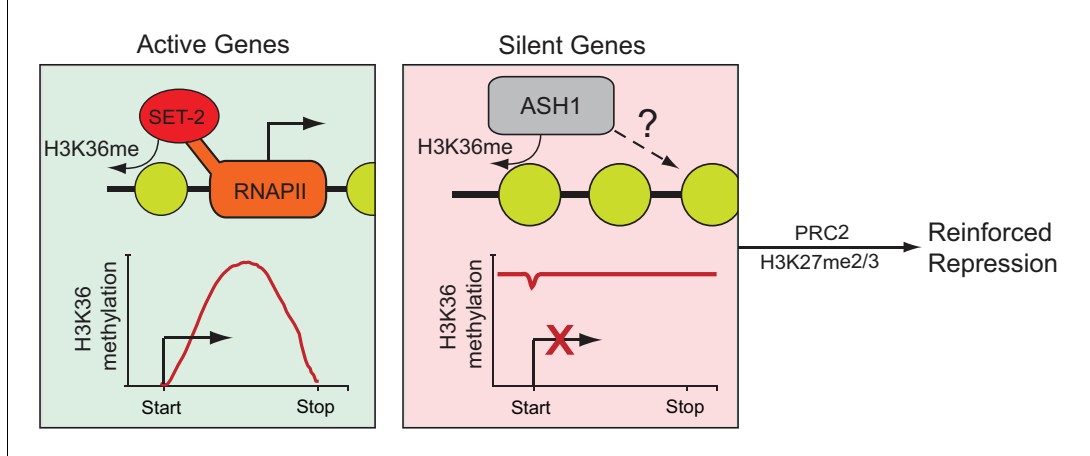

**Figure 6.** Model for H3K36me deposition in *Neurospora crassa*. Neurospora genes can be divided into two groups depending upon their source of H3K36me. SET-2 marks the gene body of actively transcribed genes, and conversion to the trimethylated state is tied to transcription. ASH1 establishes large domains of H3K36me that covers silent genes and flanking regions. ASH1 predominantly deposits H3K36me2, but also appears to have the capacity to produce H3K36me3; the significance of di- versus tri-methylation remains unclear. In both cases, H3K36me appears to have a repressive function, protecting active genes from cryptic transcription and maintaining general repression of inactive genes. Genes marked by ASH1 can be co-modified by PRC2 with H3K27me2/3. Genes marked by both ASH1 and PRC2 appear 'locked' in a dormant state (i.e., they are less likely to be activated in the absence of ASH1).
DOI: https://doi.org/10.7554/eLife.41497.013

and blocking general transcription at inactive genes. The repressed state of ASH1-modified chromatin is largely contingent upon the presence of ASH1-catalyzed H3K36me, but can be further modified with H3K27me2/3 catalyzed by the PRC2 complex to support repression.

# Materials and methods

## Key resources table

| Reagent type (species) or resource | Designation | Source or reference | Identifiers | Additional information |
|---|---|---|---|---|
| Strain, strain background (*N. crassa*) | *WT* | FGSC#6103 | N623 | *mat A; his-3* |
| Strain, strain background (*N. crassa*) | *WT* | *Honda and Selker, 2008* | N2930 | *mat A; his-3; Δmus-52::bar* |
| Strain, strain background (*N. crassa*) | *WT* | FGSC#2489 | N3752 | *mat A* |
| Strain, strain background (*N. crassa*) | *WT* | FGSC#4200 | N3753 | *mat a* |
| Strain, strain background (*N. crassa*) | *ash1-3xFLAG* | this study | N4865 | *mat A; his-3; ash1-3xFLAG::hph* |
| Strain, strain background (*N. crassa*) | *ash1(Y888F)* | this study | N4877 | *mat A; his-3; ash1(Y888F)—3xFLAG::hph; Δmus-52::bar* |
| Strain, strain background (*N. crassa*) | *ash1(Y888F)* | this study | N4878 | *mat A; his-3; ash1(Y888F)—3xFLAG::hph* |
| Strain, strain background (*N. crassa*) | *ash1(Y888F)* | this study | N6268 | *mat A; his-3; ash1(Y888F)—3xFLAG::hph* |
| Strain, strain background (*N. crassa*) | *ash1(Y888F)* | this study | N6269 | *mat A; his-3; ash1(Y888F)—3xFLAG::hph* |
| Strain, strain background (*N. crassa*) | *ash1(Y888F)* | this study | N6875 | *mat a; ash1(Y888F)—3xFLAG::nat* |
| Strain, strain background (*N. crassa*) | *ash1(Y888F)* | this study | N6878 | *mat a; ash1(Y888F)—3xFLAG::nat; Δmus-52::bar* |
| Strain, strain background (*N. crassa*) | *Δset-2* | FGSC#15504 | N5761 | *mat a; Δset-2::hph* |
| Strain, strain background (*N. crassa*) | *Δset-2* | this study | N6335 | *mat A; Δset-2::hph* |
| Strain, strain background (*N. crassa*) | *set-2(ΔSRI)* | this study | N6956 | *mat A; set-2(ΔSRI)::nat* |
| Strain, strain background (*N. crassa*) | *Δset-7* | FGSC#11182 | N4718 | *mat a; Δset-7::hph* |
| Strain, strain background (*N. crassa*) | *Δset-7* | *Jamieson et al., 2018* | N4730 | *mat A; Δset-7::bar* |

*Continued on next page*

*Continued*

| Reagent type (species) or resource | Designation | Source or reference | Identifiers | Additional information |
|---|---|---|---|---|
| Strain, strain background (*N. crassa*) | Δ*npf* | FGSC# 13915 | N4721 | *mat a; Δnpf::hph* |
| Strain, strain background (*N. crassa*) | *ash1(Y888F); Δset-2::nat* | this study | N6266 | *mat A; his-3; ash1(Y888F) −3xFLAG::hph; Δset-2::nat; Δmus-52::bar* |
| Strain, strain background (*N. crassa*) | *ash1(Y888F); Δset-2* | this study | N6267 | *mat A; his-3; ash1(Y888F) −3xFLAG::hph; Δset-2::nat* |
| Strain, strain background (*N. crassa*) | Δ*set-2; Δset-7* | this study | N6333 | *mat ?; Δ set-2::hph; Δset-7::bar* |
| Strain, strain background (*N. crassa*) | Δ*set-2; Δset-7* | this study | N6334 | *mat ?; Δset-2::hph; Δset-7::bar* |
| Strain, strain background (*N. crassa*) | csr-1: (TTAGGG)17 | *Jamieson et al., 2018* | N6383 | mat a; csr-1: (TTAGGG)17; Δmus-52::bar |
| Antibody | H3K36me3 | *Cell Signaling* | Cat#4909S, Clone (D5A7)) | immunoblot (1:1000) |
| Antibody | H3K36me2 | *Abcam* | Cat#ab9049 | immunoblot (1:1000) |
| Antibody | Histone H3 | *Abcam* | Cat#ab1791 | immunoblot (1:2000) |
| Antibody | IRDye 680RD Goat-anti-Rabbit secondary antibody | *Licor* | Cat#926–68071 | immunoblot (1:5000) |
| Antibody | H3K27me3 | *Millipore* | Cat#07–449 | Chromatin immuno precipitation |
| Antibody | H3K36me3 | *Abcam* | Cat#ab9050 | Chromatin immuno precipitation |
| Antibody | H3K36me2 | *Abcam* | Cat#ab9049 | Chromatin immuno precipitation |
| Antibody | H3K27ac | *ActiveMotif* | Cat#39133 | Chromatin immuno precipitation |
| Antibody | H3K27me2/3 | *ActiveMotif* | Cat#39535 | Chromatin immuno precipitation |

## Neurospora strains and molecular analyses

All Neurospora strains used in this study are listed in Key resources table. Strains were grown, crossed, and maintained according to standard procedures (*Davis, 2000*). Knockout and mutant strains were either taken from the Fungal Genetic Stock Center knockout collection (*Colot et al., 2006*; *McCluskey et al., 2010*) or generated as previously described (*Honda and Selker, 2009*). ASH1 mutations were made with a QuickChange site-directed mutagenesis kit (Stratagene) and PCR-based mutagenesis with the In-Fusion HD cloning system (Takara). The follow primer sets were used for quantitative real-time (qRT) PCR: *8:G3* (CGTAGAGAAGGGAAGTAGTAG; GCACAA

TACGAAGTCACTTTTCGCC), *NCU07152* (GGCAACAGAGGCTGTGCTGC, CGCAAAGA TGCCGCACCTGTC), *hH4* (CATCAAGGGGTCATTCAC, TTTGGAATCACCCTCCAG).

## Immunoblotting

Immunoblotting was performed as previously described (*Honda and Selker, 2008*). Briefly, Neurospora extracts were produced by sonication in extraction buffer (50 mM Hepes pH7.5, 1 mM EDTA, 150 mM NaCl, 10% Glycerol, 0.02% NP40) supplemented with cOmplete ULTRA protease inhibitor cocktail tablets (Roche, 05892970001). The following antibodies were used for immunoblotting: H3K36me3 (Cell Signaling, Cat#4909S, Clone (D5A7)), H3K36me2 (Abcam, Cat#ab9049), Histone H3 (Abcam, Cat#ab1791), IRDye 680RD Goat-anti-Rabbit secondary antibody (Licor, Cat#926 – 68071).

## ChIP and library preparation

ChIP was performed as previously described (*Jamieson et al., 2016*). qPCR was performed using the Quanta Biosciences PerfeCTa Sybr Green FastMix and an Applied Biosystems Step One Plus Real-Time PCR System. ChIP-libraries were prepared as previously described (*Jamieson et al., 2016*) and sequencing was performed using an Illumina NextSeq 500 or HiSeq 4000 sequencer with 75- or 100-nt single-end reads, respectively. All sequencing reads were mapped to the corrected *N. crassa* OR74A (NC12 genome) (*Galazka et al., 2016*) using Bowtie2 (*Langmead and Salzberg, 2012*). ChIP-seq read coverage was averaged, normalized, and analyzed using tools available from deepTools2 (*Ramírez et al., 2016*) and SAMtools (*Li et al., 2009*) on the open-source platform Galaxy (*Afgan et al., 2016*). Sequencing tracks are displayed as 25-nt-window TDF or bigWig files with the Integrative Genomics Viewer (IGV) (*Robinson et al., 2011*). The following antibodies were used for ChIP: H3K27me3 (Millipore, Cat#07 – 449), H3K36me3 (Abcam, Cat#ab9050), H3K36me2 (Abcam, Cat#ab9049), H3K27ac (ActiveMotif, Cat#39133), H3K27me2/3 (ActiveMotif, Cat#39535).

## RNA-seq

RNA was isolated (*Jamieson et al., 2013*), DNase treated (Thermo Fisher Scientific), cleaned (Agencourt RNAClean XP beads; Beckman Coulter), and Poly-A +RNA seq libraries prepared (KAPA Stranded mRNA-seq kit; KAPA Biosystems) and sequenced on a Illumina NextSeq 500 or HiSeq 4000 sequencer with 75- or 100-nt single-end reads, respectively. High-quality (Kmer filtering) adapter-trimmed reads were identified (Stacks) (*Catchen et al., 2013*), mapped (TopHat2) (*Kim et al., 2013*), sorted (SAMTools) (*Li et al., 2009*), and directionality-preserved read numbers for genes were calculated (HTSeq) (*Anders et al., 2015*). Differential gene expression (DESeq2) (*Love et al., 2014*) analysis examined pair-wise differences between WT and mutants or within replicates.

## Sequencing analysis and bioinformatics

Sequencing analysis was performed with previously described software using the open-source platform Galaxy (*Afgan et al., 2016*). Tools available from DeepTools (*Ramírez et al., 2016*) were used for the following: 1) bamCoverage was used to generate coverage bigWig files; 2) bamCompare was used to normalize and obtain $log_2$ratios from two BAM files; 3) computeMatrix was used to prepare data for heatmaps or profiles; 4) plotHeatmap was used to create heatmaps for score distributions; 5) plotProfile was used to create meta-plots of score distributions. Tools available from SAMtools (*Li et al., 2009*) were used for the following: 1) BedCov was used to calculate read depth over given intervals; 2) Merge BAM Files was used to combine replicates. GraphPad Prism was used to analyze frequencies and prepare histograms.

## Accession numbers

Complete ChIP-seq and RNA-seq files, gene expression values, ChIP-seq intensity values have been deposited in NCBI's Gene Expression Omnibus (GEO; http://ncbi.nlm.nih.gov/geo) and are accessible through GEO Series accession number GSE118495 and, as part of a previously reported series GSE82222 (*Klocko et al., 2016*) and GSE104019 (*Jamieson et al., 2018*).

## Materials
Requests for materials should be addressed to VTB and EUS. All *Neurospora crassa* strains are available at the Fungal Genetic Stock Center (*McCluskey et al., 2010*).

# Acknowledgements
The authors thank the Genomics Core Facility at the University of Oregon for carrying out the high-throughput DNA sequencing, A Klocko, E Wiles, and K McNaught for carrying out related exploratory experiments, A Maguire, H Duvvuri, J Smith, H Manning, and D Diaz for carrying out related bioinformatics experiments, and J McKnight, E Wiles, and K McNaught for comments on the manuscript. This work was funded by grants from the National Institutes of Health to VTB (CA180468) and EUS (GM093061, GM035690 and GM127142).

# Additional information

### Funding

| Funder | Grant reference number | Author |
| --- | --- | --- |
| National Institutes of Health | CA180468 | Vincent T Bicocca |
| National Institutes of Health | GM093061 | Eric U Selker |
| National Institutes of Health | GM035690 | Eric U Selker |
| National Institutes of Health | GM127142 | Eric U Selker |

The funders had no role in study design, data collection and interpretation, or the decision to submit the work for publication.

### Author contributions
Vincent T Bicocca, Conceptualization, Data curation, Formal analysis, Funding acquisition, Validation, Investigation, Visualization, Methodology, Writing—original draft, Writing—review and editing; Tereza Ormsby, Investigation, Writing—review and editing; Keyur K Adhvaryu, Shinji Honda, Investigation; Eric U Selker, Conceptualization, Resources, Supervision, Funding acquisition, Project administration, Writing—review and editing

### Author ORCIDs
Vincent T Bicocca (iD) http://orcid.org/0000-0002-5702-4586
Eric U Selker (iD) http://orcid.org/0000-0001-6465-0094

### Decision letter and Author response
Decision letter https://doi.org/10.7554/eLife.41497.022
Author response https://doi.org/10.7554/eLife.41497.023

# Additional files

### Supplementary files
• Transparent reporting form
DOI: https://doi.org/10.7554/eLife.41497.014

### Data availability
All source data files can be found with our GEO submission (GSE118495). The accession numbers for our GEO data set and the data sets of other relevant submissions are included in the Materials and Methods section. GEO submissions include raw HT-sequencing files for all biological replicates, the processed version of these files ready for display on IGV (or other viewer), associated data tables, and region/domain definitions (BED files).

The following dataset was generated:

| Author(s) | Year | Dataset title | Dataset URL | Database and Identifier |
|---|---|---|---|---|
| Bicocca VT, Orms-by T, Adhvaryu KK, Honda S, Selker EU | 2018 | ASH-1-catalyzed H3K36 methylation drives gene repression and marks H3K27me2/3-competent chromatin | https://www.ncbi.nlm.nih.gov/geo/query/acc.cgi?acc=GSE118495 | NCBI Gene Expression Omnibus, GSE118495 |

The following previously published datasets were used:

| Author(s) | Year | Dataset title | Dataset URL | Database and Identifier |
|---|---|---|---|---|
| Klocko AD, Ormsby T, Galazka JM, Uesaka M, Leggett N, Honda S, Frei-tag M, Selker EU | 2016 | Neurospora crassa genome organization requires subtelomeric facultative heterochromatin | https://www.ncbi.nlm.nih.gov/geo/query/acc.cgi?acc=GSE82222 | NCBI Gene Expression Omnibus, GSE82222 |
| Jamieson K, McNaught KJ, Leggett NA, Orms-by T, Honda S, Selker EU | 2018 | Telomere repeats induce domains of H3K27 methylation in Neurospora | https://www.ncbi.nlm.nih.gov/geo/query/acc.cgi?acc=GSE104019 | NCBI Gene Expression Omnibus, GSE104019 |

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
