## [Decision Letter]

Thank you for submitting your article "ASH1-catalyzed H3K36 methylation drives gene repression and marks H3K27me2/3-competent chromatin" for consideration by *eLife*. Your article has been reviewed by three peer reviewers, and the evaluation has been overseen by Jerry Workman as Reviewing Editor and Kevin Struhl as the Senior Editor. The reviewers have opted to remain anonymous.

The reviewers have discussed the reviews with one another and the Reviewing Editor has drafted this decision to help you prepare a revised submission.

Summary:

The study by Bicocca et al. investigates the role of H3K36 methylation and its interplay with H3K27 methylation in *Neurospora*. The study starts out by providing evidence that in *Neurospora*, the bulk of H3K36me3 and substantial amounts of H3K36me2 are generated by SET-2 and that ASH1 is a second HMTase that is making a major contribution to generating the bulk of H3K36me2. The authors go on to show that SET-2 and ASH1 contribute to the genome-wide H3K36me2 and -me3 profiles in a locus-specific manner. RNAseq analyses unexpectedly reveal that among the genes that are misregulated in *ash1* mutants, a substantially larger fraction of genes is upregulated compared to down-regulated genes. It appears that ASH1 is required for repression of poorly transcribed genes. The authors show that about 30% of the genes that are marked with H3K36me2 by ASH1 are also decorated with H3K27me2/3 generated by PRC2. Surprisingly, about 30% of the genes that are co-marked with H3K36me2 and H3K27me2/3 show strong reduction or complete loss of H3K27me2/3, gain of H3K27ac and up-regulated expression of RNA. Nevertheless, the authors also find regions where the loss of H3K36me2 in *ash1* mutants is accompanied by a marked gain of H3K27me2/3 (Figure 5F), consistent with the 'conventional' view that H3K36me2/3 directly antagonizes H3K27 methylation by PRC2. The authors conclude that ASH1 catalyzed H3K36me2 can affect H3K27me2/3 accumulation both in a positive and in a negative manner.

This study is interesting. At some genomic regions, H3K36me2/3 deposited by ASH1 antagonizes H3K27 methylation by PRC2, consistent with previous findings made in *Drosophila* (Papp and Müller, 2006; Srinivasan et al., 2008; Dorighi et al., 2013), in worms (Gaydos et al., 2012) and in experiments that used purified PRC2 for HMTase reactions on nucleosomes in vitro (Schmitges et al., 2011; Yuan et al., 2011). On the other hand, the authors also present data that stand in direct contrast to this regulatory relationship. Specifically, the ChIP analyses suggest that ASH1-mediated H3K36me2 deposition can – somehow – act in a positive way to promote H3K27 methylation by PRC2. The authors discuss these findings that are not so simple to reconcile in a balanced way. Perhaps, their discussion could also touch on a two main issues mentioned in point 4 below.

Essential revisions:

1) Looking at WB in Figure 1E, it seems that there is still a very faint H3K36me3 WB signal in lane 7 and that this signal is comparable to that seen in lanes 3 and 4 of the same blot. So, from this, it does not look like ASH1 would contribute to generating H3K36me3, whereas in lane 4 (*set-2 ash1* double mutant) in Figure 1F it indeed appears that there is a further reduction of H3K36me3 signal compared to lane 3 (*set-2* single mutant). The authors should clarify this point because, as the authors point out in the Introduction, there is no evidence that HMTases of the ASH1/NSD class are at all capable to tri-methylate H3K36. Since the effects are subtle, it would have been useful to show a more quantitative analysis of bulk modification levels, by performing western blots on serial dilutions (e.g. 4:2:1) of the extracts, rather than just one amount of extract per genotype. If the authors have more quantitative WB data, we encourage them to add them to the manuscript.

2) Figure 2. First, how were the ChIP-seq reads from wt, *ash1* and *set-2* mutants normalized to take into account the changes in bulk H3K36me2 and -me3 levels? Second, have the authors generated H3K36me2 and -me3 profiles in *set-2 ash1* double mutant cells?

3) For the RNA-seq analysis, it would be good to have more quantitative information on the extent of up- and down-regulation. In particular, how many genes are more than 2-fold up- or down-regulated and how many genes are more than 4-fold up- or down-regulated?

4) These are two points that the authors may want to discuss.

First, what is the function of H3K27me2/3 in *Neurospora*? Does *Neurospora* contain a chromodomain-containing protein in a PRC1-type complex (i.e. like Pc/CBX2 in PRC1 in animals)?

The second point that might be interesting to discuss is that unlike in flies and mammals, in plants, not all forms of PRC2 are inhibited by active marks (Schmitges et al., 2011). Specifically, PRC2 complexes containing the Su(z)12 ortholog EMF2 are inhibited by H3K4me3 but PRC2 containing the Su(z)12 ortholog VRN2 are not inhibited by this mark (Schmitges et al., 2011). So, it remains to be tested whether the HMTase activity of *Neurospora* PRC2 is at all inhibited on H3K36me2/3 nucleosomes in vitro.

---

## [Author Response]

Essential revisions:1) Looking at WB in Figure 1E, it seems that there is still a very faint H3K36me3 WB signal in lane 7 and that this signal is comparable to that seen in lanes 3 and 4 of the same blot. So, from this, it does not look like ASH1 would contribute to generating H3K36me3, whereas in lane 4 (set-2 ash1 double mutant) in Figure 1F it indeed appears that there is a further reduction of H3K36me3 signal compared to lane 3 (set-2 single mutant). The authors should clarify this point because, as the authors point out in the Introduction, there is no evidence that HMTases of the ASH1/NSD class are at all capable to tri-methylate H3K36. Since the effects are subtle, it would have been useful to show a more quantitative analysis of bulk modification levels, by performing western blots on serial dilutions (e.g. 4:2:1) of the extracts, rather than just one amount of extract per genotype. If the authors have more quantitative WB data, we encourage them to add them to the manuscript.

We share the concern regarding the best way to demonstrate the subtle change between the *set-2* mutant and the *ash1/set-2* double mutant. The contribution of ASH1 is more obvious in Figure 1F (perhaps helped by the lanes being directly adjacent) than in Figure 1E, but we believe the conclusions drawn from both blots are consistent. In each case, the signal remaining after *set-2* deletion is very faint, suggesting SET-2 is responsible for nearly all of H3K36me3. Though faint, the intensity of this remaining signal is reduced when combined with the *ash1* mutant. The phenotype is robust in its reproducibility, but there is slight variation in the apparent contribution of ASH1. We felt these blots were representative of the phenotype, but took the recommendation of performing serial dilution blots to further demonstrate the phenotype. The new data, now shown as Figure 1F, reinforce our conclusion.

Regarding evidence of H3K36me3 activity from ASH1/NSD orthologs, we note in the Introduction that no in vitro analysis of ASH1 orthologs has demonstrated H3K36me3 activity. However, as we note in the Discussion section, there is in vivo evidence for it. Janebska et al. (2017) and Jiang et al. (2013) describe H3K36me3 activity from ASH1-orthologs in *Fusarium fujikuroi* and *Plasmodium falciparum*. In addition, *C. elegans mes-4* (an ortholog of NSD1) has been reported to have H3K36me3 activity (Furuhashi et al., 2010).

2) Figure 2. First, how were the ChIP-seq reads from wt, ash1 and set-2 mutants normalized to take into account the changes in bulk H3K36me2 and -me3 levels? Second, have the authors generated H3K36me2 and -me3 profiles in set-2 ash1 double mutant cells?

ChIP-seq tracks are shown with as little manipulation as possible; they are simply normalized to reads per kilobase per million (RPKM) and averaged over 25bp bins. We do not think the ChIP-seq tracks should be used to interpret the relative intensity of H3K36me catalyzed by SET-2 versus ASH1 and tried not to make claims to suggest they could. Largely, we think each mutant should be considered independently to draw more qualitative conclusions about where the mark is being found. However, the data do provide an opportunity to compare the relative contribution of SET-2 and ASH1 to the profile observed in WT. Consider the 50kb region we highlight in Figure 2A and 2C, where we see that SET-2 contributes little or no H3K36me2 signal, but in the WT track we still observe a H3K36me2 signal that is comparable in intensity to adjacent regions. This suggests ASH1 is both responsible for nearly all the H3K36me2 signal found at this location and the intensity of the ASH1 signal is comparable to that catalyzed by SET-2 in the neighboring region.

We were able to build a *set-2/ash1* double mutant. As one might expect, ChIP of H3K36me from the *set-2/ash1* double mutant yielded very little DNA (less than what we typically need to generate libraries) and the ChIP-seq tracks we interpreted as non-specific. The *set-2/ ash1* double mutant was more valuable as a control in ChIP qRT-PRC experiments, which avoid some of the normalization concerns of ChIP-seq experiments. We have therefore added qRT-PCR results to Figure 2—figure supplement 1 to better demonstrate the relative signal contribution from ASH1 and SET-2, and to show that the signal attributed to either ASH1 and SET-2 is eliminated in the double mutant.

H3K36me3 ChIP results are shown for WT, Δ*set-2*, and Δ*set-2; ASH1*(Y888F) strains at three genomic regions with distinct H3K36me profiles. 8:G3 is a constitutive heterochromatin region that lacks H3K36me and is used to assess “background” in the ChIP. *hH4* is an actively expressed gene that is marked by SET-2-catalyzed H3K36me but not ASH1-catalyzed H3K36me. *NCU07152* in a silent uncharacterized gene that is densely marked by ASH1-catalyzed H3K36me and H3K27me2/3.

3) For the RNA-seq analysis, it would be good to have more quantitative information on the extent of up- and down-regulation. In particular, how many genes are more than 2-fold up- or down-regulated and how many genes are more than 4-fold up- or down-regulated?

We agree and have included a more quantitative analysis of our RNA-seq data in a table included as Figure 5—figure supplement 2. Δ*set-7* data were taken from Klocko et al. (2016).

4) These are two points that the authors may want to discuss.First, what is the function of H3K27me2/3 in Neurospora? Does Neurospora contain a chromodomain-containing protein in a PRC1-type complex (i.e. like Pc/CBX2 in PRC1 in animals)?

Although it is clear that PRC2 is repressive in *Neurospora* (numerous genes are derepressed when complex components are eliminated genetically), the mechanism remains unknown. We revised the Discussion section to provide additional background and context for H3K27me2/3 in *Neurospora* and we have specifically addressed conservation (or lack thereof) of the PRC2 and PRC1 complexes. Briefly, as in some other organisms, no obvious homologs of the PRC1 components are apparent in the *Neurospora* genome. However, it is clear that the presence of H3K27me2/3 is associated with repression (Wiles and Selker, 2016). Ongoing work is directed at discovering the mechanism behind H3K27me2/3-associated repression in *Neurospora*, and the mechanism of repression by H3K27me2/3 is not completely clear even in organisms sporting obvious PRC1 homologs. Moreover, recent work has called into question seeming established models for the role of PRC1 in repression (King et al., 2018).

The second point that might be interesting to discuss is that unlike in flies and mammals, in plants, not all forms of PRC2 are inhibited by active marks (Schmitges et al., 2011). Specifically, PRC2 complexes containing the Su(z)12 ortholog EMF2 are inhibited by H3K4me3 but PRC2 containing the Su(z)12 ortholog VRN2 are not inhibited by this mark (Schmitges et al., 2011). So it remains to be tested whether the HMTase activity of Neurospora PRC2 is at all inhibited on H3K36me2/3 nucleosomes in vitro.

Schmitges et al. (2011) is an interesting paper and may be relevant to the observations we report. Although not known in *Neurospora*, PRC2 complexes that consist of different components could potentially explain why only certain H3K27me2/3 genes are derepressed when ASH1 is inactivated. Of possible relevance, earlier work from our lab showed partial loss of H3K27me2/3 associated with deletion of the gene encoding *Neurospora* p55 (NPF; Jamieson et al., 2013), which is thought to be a component of *Neurospora* PRC2. The mutant specifically loses H3K27me2/3 at telomere-proximal regions. This is in stark contrast to the ASH1 mutant, which retains telomere-proximal regions. In fact, when we categorize H3K27me2/3-marked genes into NPF-dependent and ASH1-dependent we see they each influence a similar number of genes (about 30% of all H3K27me2/3-marked genes). If NPF and ASH1 were randomly regulating H3K27me2/3 genes we would expect 30% of these genes would be regulated by both NPF and ASH1; however, we see only 17% overlap, suggesting they are actively regulating separate compartments of H3K27me2/3-marked genes.